# Measurements of Anti-SARS-CoV-2 Antibody Levels after Vaccination Using a SH-SAW Biosensor

**DOI:** 10.3390/bios12080599

**Published:** 2022-08-04

**Authors:** Chia-Hsuan Cheng, Yu-Chi Peng, Shu-Min Lin, Hiromi Yatsuda, Szu-Heng Liu, Shih-Jen Liu, Chen-Yen Kuo, Robert Y. L. Wang

**Affiliations:** 1Tst Biomedical Electronics Co., Ltd., Taoyuan 324403, Taiwan; 2Biotechnology Industry Master and Ph.D. Program, Chang Gung University, Taoyuan 33302, Taiwan; 3Department of Thoracic Medicine, Chang Gung Memorial Hospital, Linkou 33305, Taiwan; 4National Institute of Infectious Diseases and Vaccinology, Taoyuan 33302, Taiwan; 5Division of Pediatric Infectious Diseases, Department of Pediatrics, Chang Gung Memorial and Children’s Hospital, Linkou 33305, Taiwan; 6Department of Biomedical Sciences, College of Medicine, Chang Gung University, Taoyuan 33302, Taiwan; 7Kidney Research Center and Department of Nephrology, Chang Gung Memorial Hospital, Linkou 33305, Taiwan

**Keywords:** SARS-CoV-2, SH-SAW biosensor, vaccine, antibody

## Abstract

To prevent the COVID-19 pandemic that threatens human health, vaccination has become a useful and necessary tool in the response to the pandemic. The vaccine not only induces antibodies in the body, but may also cause adverse effects such as fatigue, muscle pain, blood clots, and myocarditis, especially in patients with chronic disease. To reduce unnecessary vaccinations, it is becoming increasingly important to monitor the amount of anti-SARS-CoV-2 S protein antibodies prior to vaccination. A novel SH-SAW biosensor, coated with SARS-CoV-2 spike protein, can help quantify the amount of anti-SARS-CoV-2 S protein antibodies with 5 μL of finger blood within 40 s. The LoD of the spike-protein-coated SAW biosensor was determined to be 41.91 BAU/mL, and the cut-off point was determined to be 50 BAU/mL (Youden’s J statistic = 0.94733). By using the SH-SAW biosensor, we found that the total anti-SARS-CoV-2 S protein antibody concentrations spiked 10–14 days after the first vaccination (*p* = 0.0002) and 7–9 days after the second vaccination (*p* = 0.0116). Furthermore, mRNA vaccines, such as Moderna or BNT, could achieve higher concentrations of total anti-SARS-CoV-2 S protein antibodies compared with adenovirus vaccine, AZ (*p* < 0.0001). SH-SAW sensors in vitro diagnostic systems are a simple and powerful technology to investigate the local prevalence of COVID-19.

## 1. Introduction

Severe acute respiratory syndrome coronavirus 2 (SARS-CoV-2) is the cause of the 2019 coronavirus (COVID-19) pandemic [1]. It is an enveloped, positive-sense, and single-stranded RNA virus, which is transmitted through surface contamination, aerosol, and the fecal-oral route. The SARS-CoV-2 genome encodes four structural proteins including nucleocapsid (N) protein, spike (S) protein, membrane (M) protein, and envelop (E) protein [2,3]. The S and N proteins are the most immunogenic proteins of SARS-CoV-2 [4]. Recently, there are several COVID-19 vaccines in development for pandemic control and prevention of COVID-19. The vaccines include adenoviral-vectored vaccines, nucleic acid vaccines, subunit protein vaccines, and whole-cell inactivated virus vaccines [5,6]. The available COVID-19 vaccines in Taiwan are ChAdOx1 nCoV-19 by AstraZeneca/Oxford, BNT-162b2 by BioNTech/Pfizer, mRNA-1273 by Moderna, and MVC-COV1901 by Medigen. ChAdOx1 nCoV-19 is the adenoviral-vectored vaccine. Both BNT-162b2 and mRNA-1273 are mRNA vaccines. MVC-COV1901 is a recombinant protein subunit vaccine [6,7].

Follow-up measurements of anti-SARS-CoV-2 antibodies after vaccination are a potential surrogate marker of protection [5,8]. The general anti-SARS-CoV-2 antibody detection methods include enzyme-linked immunosorbent assay (ELISA) and lateral flow immunoassay (LFIA). ELISA is characterized by high throughput and high sensitivity. It needs to be performed in a conditioned laboratory due to the complexity of the operation steps. The measurement time of LFIA is short and some interfering factors may lead to false positive results [9,10]. Surface acoustic wave (SAW) biosensors have been used to detect various biomarkers with high selectivity, high sensitivity, and low cost [11,12]. Our previous study showed that shear-horizontal surface acoustic wave (SH-SAW) biosensors can measure anti-SARS-CoV-2 N protein antibodies with good sensitivity [13]. The SH-SAW biosensor contains an input and output interdigital transducer (IDT), a sensing area coated with antibodies or antigens of the target molecules, and a reflector. When the measurement starts, the incoming electrical signal is converted into surface acoustic waves by the input IDT. The wave then propagates along the sensing zone and is reflected through the reflector to the output IDT. Finally, the wave is converted into an electrical signal. The output signal is associated with the binding of the sensing zone. As the target molecules bind to the sensing region, the velocity and amplitude of the wave attenuate. The more the target molecules bind to the sensing region, the larger the signal it produces [12,14,15].

In this study, we demonstrated the performance of an S-protein coated SH-SAW biosensor and established a 4PL curve. Afterwards, we compared the antibody responses to each vaccine (AstraZeneca/Oxford, BioNTech/Pfizer, Moderna, and Medigen). In addition, we investigated the kinetic properties of total anti-SARS-CoV-2 S protein antibody levels in whole blood after vaccination. Our study showed that the concentrations of total anti-SARS-CoV-2 S protein antibodies spiked 10–14 days after the first vaccination. The concentration of total anti-SARS-CoV-2 S protein antibodies could reach 322.9 ± 614.4 BAU/mL at 29–56 days after the first vaccination. The concentration of total anti-SARS-CoV-2 S protein antibodies spiked and peaked 7–9 days after the second vaccination. This SH-SAW sensor system can easily confirm the amount of anti-S antibodies before additional injections of COVID-19 vaccine. This must be very effective for additional vaccination decisions.

## 2. Materials and Methods

### 2.1. Materials

iProtin immunoassay reader and SH-SAW biosensor chips were supplied by tst biomedical electronics Co., Ltd. (Taoyuan, Taiwan). Hellmanex III (259304) was purchased from Hellma Analytics (Munich, Germany). Dithiobis [succinimidyl propionate] (DSP) (VI309258) and dimethyl sulfoxide (DMSO) (TH270381) were obtained from Thermo Fisher Scientific (Waltham, MA, USA). The SARS-CoV-2 trimeric S protein coated on the capture channel of the SH-SAW biosensor chip was from the National Health Research Institutes. Casein (97062-926) was purchased from VWR (Radnor, PA, USA). Phosphate buffered saline (CWFF0613) and bovine albumin serum (FUBSA001.100) were bought from Bio-Future company (BioFuture biotech, Taoyuan, Taiwan). The first WHO International Standard for anti-SARS-CoV-2 immunoglobulin (human) (NIBSC code: 20/136) was purchased from the National Institute for Biological Standards and Control. The stabilizer was prepared by adding 15 µL of Tween 20 to 10% sucrose (BCBV9208, Sigma, St. Louis, MO, USA).

In this study, we used a 3 mm × 5 mm dual-channel SH-SAW biosensor chip; one channel for reference and the other for capture. Each channel on the chip has an IDT, a reflector, and a sensing zone between them. The dual-channel SH-SW biosensor chip mounted on a printed circuit board (PCB) is shown in Figure 1.

### 2.2. Fabrication of SH-SAW Biosensor Chips Coated with S Protein

First, the sensing area of the SH-SAW biosensor chip was cleaned with O_2_ plasma for 10 min. 2% of Hellmanex III was added and incubated for 20 min. Then, the chip was rinsed twice with double distilled water. We then added 0.4 mg/mL of DSP solution (in DMSO) and incubated for 20 min. After that, the chip was rinsed with DMSO. The chip was then washed with double distilled water and air dried. The reference and capture channels were coated with 10% bovine albumin serum (in double distilled water) and 0.45 mg/mL of SARS-CoV-2 trimeric S protein, respectively. After that, the chips were blocked with 2% casein pH 7.4 (in PBS). Finally, stabilizer was added and the chips were blown dry. The chips were labeled and stored in the dry carbinet.

### 2.3. Establishment of the 4PL Standard Curve

The WHO International Standard for anti-SARS-CoV-2 immunoglobulin (human) was used as the standard. The standard was serially diluted and mixed with whole blood. The standards were measured and a 4PL standard curve was established; the equation for the 4PL standard curve is as follows:$$y=d+\frac{a-d}{1+{\left(\frac{x}{c}\right)}^{b}}$$
where a, b, c, and d are coefficients. X is the titre of anti-SARS-CoV-2 S protein total antibody. Anti-SARS-CoV-2 S protein total antibody levels were quantified in BAU/mL.

### 2.4. Measurement of Total Anti-S Protein Antibody Using SH-SAW Biosensor

Total anti-SARS-CoV-2 trimeric S protein antibodies were measured by iProtin immunoassay and SARS-CoV-2 S protein-coated SH-SAW sensor chips (Figure 2). After the iProtin immunoassay was pre-warmed, the QR code containing the 4PL standard curve was scanned. The SARS-CoV-2 S protein assay cassette was inserted into the slot of the iProtin immunoassay reader. Then, 5 µL of whole blood was dropped onto the chip. After 3 min of incubation, the antibody level was displayed on the screen of the iProtin immunoassay reader.

### 2.5. Clinical Trial

Participants in this study included subjects who had received a SARS-CoV-2 vaccine. The study protocol was reviewed and approved by the Institutional Review Board (IRB2108130013). Inclusion criteria were adults between the ages of 20 and 65 years. Subjects diagnosed with metabolic disorders or immunodeficiency diseases were excluded. Whole blood was collected from participants who consented to participate in this study.

### 2.6. Statistical Analysis

Quantitative data were expressed as means ± standard deviation (SD) and compared using the Student’s *t*-test. A *p*-value < 0.05 was considered statistically significant. Statistical analyses were performed using SPSS statistics 17 (SPSS, Chicago, IL, USA).

## 3. Results

### 3.1. Characterization of S Protein Coated SAW Biosensor

To establish the standard curve, the SAW signals for different concentrations of total anti-S protein antibodies are plotted in Figure 3 and fitted with the following four-parameter logistics (4PL) equation (Figure 4):SAW signal = A + (B − A)/{1 + ([Anti-S tAb]/C)^D}
where A = 0.9199, B = 0.0255, C = 1066.4, and D = 1.5054 are the coefficients for the SAW biosensor chip with a coefficient of correlation (R) of 0.9947; [Anti-S tAb] is the concentration of the first WHO international standard for anti-SARS-CoV-2 immunoglobulin (human).

### 3.2. Performance of S Protein Coated SAW Biosensor

To determine the limit of detection (LoD) of the S protein-coated SAW biosensor, the LoD was determined by the mean (Mean_BLK_) of 60 blank samples plus 1.645 times (σ) the standard deviation of the 60 lowest concentrations, with the following equation:LoD *=* Mean_BLK_ + 1.645 *σ_lowest conce_*.

The LoD of the S protein-coated SAW biosensor was determined to be 41.91 BAU/mL. From a linearity study, the S protein-coated SAW biosensor was tested between 48.3 to 1597.2 BAU/mL, with a linear regression of 0.9935. According to the receiver operating characteristic (ROC) analysis shown in Figure 5, when the cut-off point was determined to be 50 BAU/mL, the Youden’s J statistic was 0.94733.

### 3.3. Epidemiologic Surveillance of the Total Anti-SARS-CoV-2 S Protein Antibodies after Vaccination

A total of 25 subjects participated in the follow-up measurements of total anti-SARS-CoV-2 S protein antibodies after vaccination from July 2021 to February 2022, as shown in Figure 6. Among them, 16 subjects received the AZ vaccine, 4 subjects received BNT, 1 subject received Medigen, and 4 subjects received Moderna in the first vaccination; 10 subjects received AZ, 4 subjects received BNT, 1 subject received Medigen, and 5 subjects received Moderna in the second vaccination. Subsequently, 2 subjects received a third dose of Medigen vaccine, and 5 subjects received a third dose of Moderna vaccine in the third vaccination.

### 3.4. Quantitative Analysis of the Total Anti-SARS-CoV-2 S Protein Antibodies

The concentrations of total anti-SARS-CoV-2 S protein antibodies after vaccination with different brands of vaccines were analyzed (Figure 7). First, we collected and measured the antibodies after the vaccination, regardless of the brand, and found that the concentrations of total anti-SARS-CoV-2 S protein antibodies in the second vaccination samples were much higher than the first vaccination (Figure 7a,b). Interestingly, we found that mRNA vaccines, such as Moderna or BNT, could achieve higher concentrations of total anti-SARS-CoV-2 S protein antibodies than the adenovirus vaccine, AZ (*p* < 0.0001) (Figure 7c). Moreover, the concentrations of total anti-SARS-CoV-2 S protein antibodies could reach more than 3000.5 ± 0.71 BAU/mL after two BNT vaccinations, which was higher than that of two Moderna vaccinations (1708.5 ± 632.39 BAU/mL, *p* = 0.03) (Figure 7d).

### 3.5. Follow-Up Measurements of Total Anti-SARS-CoV-2 S Protein Antibodies after Vaccination

In all vaccinated subjects, the concentrations of total anti-SARS-CoV-2 S protein antibodies spiked 10–14 days after the first vaccination (*p* = 0.0002, compared with pre-inoculation antibody concentrations) (Figure 8a). The concentration of total anti-SARS-CoV-2 S protein antibodies could reach 322.9 ± 614.4 BAU/mL at 29–56 days after the first vaccination (Figure 8a). The concentration of total anti-SARS-CoV-2 S protein antibodies spiked and peaked 7–9 days after the second vaccination (*p* = 0.0116, compared with the pr-inoculation antibody concentration) (Figure 8b).

## 4. Discussion

The SH-SAW biosensor measures the phase shifts of the output signal based on binding event on the sensing region. After dropping the sample, the target antibody in the sample is bound to the S proteins on the surface of the chip. Since all anti-S antibodies, such as IgA, IgM, and IgG, bind to S proteins on the surface, the SH-SAW biosensor detects total anti-S antibodies. To establish the calibration curve, the WHO international SARS-CoV-2 antibody standard (20/136) was used. The electrical phase shifts of the output signal were measured 3 min after dropping the sample. The data were analyzed. As shown in Figure 3, for the range of concentrations to be measured, the concentration can be estimated by the time rate of the phase shifts for tens of seconds after the sample drop. We confirmed that a 40 s measurement time is good enough and have performed a number of experiments using a 40 s measurement time.

This study showed the kinetic properties of total anti-SARS-CoV-2 S protein antibodies after vaccination. The results showed that total anti-SARS-CoV-2 S protein antibodies spiked on days 10 to 14 after the first dose, and on days 7 to 9 after the second dose. For the mRNA vaccines, some studies showed that the vaccine-induced antibody production occurred within 2 weeks after the first dose. Additionally, antibody levels increase to a peak within a week after the second dose [1,5]. Anti-SARS-CoV-2 S protein antibodies were detectable 14 days after the first dose of AZ vaccine and peaked at 28 days. Additionally, antibody levels increase to a peak at 14 days after the second dose of AZ vaccine [6,16].

The results of this study also suggested that there are individual differences in the antibody response to vaccination. Potential factors are age, gender, and genetics [17]. Several studies have shown that women and men have an overall similar antibody response after vaccination, however some women have a faster decrease in antibody levels. In contrast, women have an overall stronger immune response after vaccination than men [18,19]. The antibody responses after vaccination were lower and antibody levels decreased more rapidly in older adults. A study showed that there were significant differences in vaccine-initiated antibody responses between older and younger people after BNT vaccination [19,20].

The SH-SAW biosensor-based anti-SARS-CoV-2 S protein antibody assay has many advantages. First, the operation procedure is simple. No labeling and washing process is required. Furthermore, the SAW platform assay does not require secondary antibodies or antibody-conjugated gold nanoparticles to enhance the signal. Secondly, it takes only forty seconds to achieve quantitative results, which is much faster than other immunoassays such as ELISA or high sensitivity chemiluminescence enzyme immunoassay (HISCL) [21]. Forty seconds of measurement time constitutes a great benefit for practical applications in clinics. Third, the exact volume of the sample is not required, and the sample can be whole blood, plasma, serum, or saliva [22]. Even for whole blood samples with blood cells and many different proteins, the binding event between the capture protein causes a change in the SAW velocity and the concentration estimated. On our SH-SAW biosensor platform, a physician can apply around 5 micro-liters of finger-pricked blood on the sensor chip and then obtain a quantitative result after 40 s. Compared to electrochemistry methods, it greatly simplifies the procedure and avoids the pain of venipuncture [23]. Finally, the SH-SAW biosensor can measure antibodies against SARS-CoV-2 total S protein. Much has been reported about the increase of IgA, IgM, IgG, and other antibody levels after SARS-CoV-2 infection. During the infection phase, the dynamics of the different antibody isotypes are different. Over time, IgG becomes dominant compared with IgA and IgM [24]. When total antibody levels are measured, it can be assumed that they are almost exclusively IgG antibody levels. Total anti-S antibodies measurement is valid for true commercial use.

Neutralization levels were highly predictive of immune protection, with 50% protection at a neutralization level equivalent to 20.2% of the mean convalescent level [25]. Moreover, the presence of RBD- or S protein-directed antibodies correlated with the level of neutralization of SARS-CoV-2, which strongly suggests that monitoring of anti-RBD and anti-S protein antibody responses could be a reliable and high-throughput compatible alternative to neutralization assays [26]. Therefore, a reliable quantitative method of SARS-CoV-2 antibody detection is needed to identify possible vaccine failure and estimate the duration of protection. The SAW platform can be used as a rapid and convenient method to determine vaccine immune protection and to help control future epidemiological trajectories.

Although we now have access to a variety of quantitative assays with CE marked antibodies against viral spike proteins, their results are not interchangeable, even when converted to BAU per milliliter using the NIBSC WHO international standard for SARS-CoV-2 immunoglobulin [27]. It is better to track one’s antibody levels with the same quantitative assay, and the palm-sized iProtin immunoassay reader is a good device for long-term monitoring of the number of antibodies after vaccination. To date, all good quantitative assays of relevance are for informational purposes only, the assay still needs to provide a cut-off point for determining the presence of antibodies to the SARS-CoV-2 S protein.

We are now developing a two-in-one kit for COVID anti-S and anti-N antibodies. The test kit has three channels; the first channel is coated with S proteins, the second channel is coated with N proteins, and the third channel is coated with blocking proteins. The third channel is used as a reference. The test kit informs us whether we are infected or not, as well as the amount of the anti-S antibodies at the same time. It is very useful in commercial applications.

## 5. Conclusions

It is important for us to know the amount of anti-SARS-CoV-2 S protein antibodies produced after vaccination, and it is also important to know if we have been infected. If we measure anti-SARS-CoV-2 N protein antibodies, we know whether we have previously been infected. The SH-SAW sensor system is useful to investigate the local prevalence of COVID-19. The simple test procedure of placing a drop of whole blood from the fingertip on the test kit, and then getting the results after 40 s provides more opportunities to obtain big data. In addition, we can easily confirm the amount of anti-S and anti-N antibodies before additional booster injections of COVID-19 vaccines. This information is very effective for additional vaccination decisions.

## Figures and Tables

**Figure 1 biosensors-12-00599-f001:**
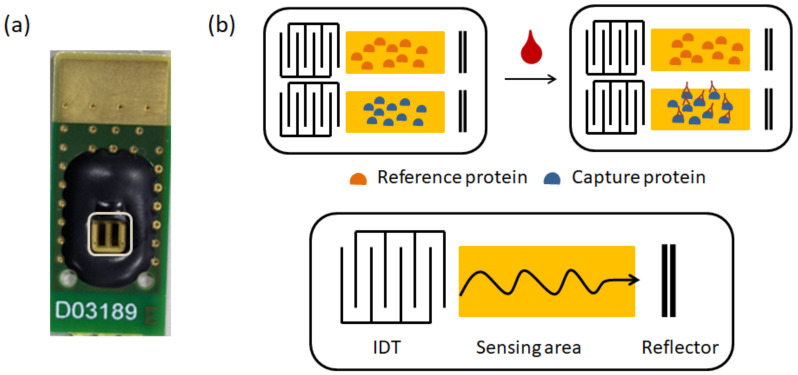
The dual-channel SH-SAW sensor chip. (**a**) Photograph of SH-SAW sensor chip; (**b**) schematic coated proteins in the dual-channel SH-SAW sensor chip.

**Figure 2 biosensors-12-00599-f002:**
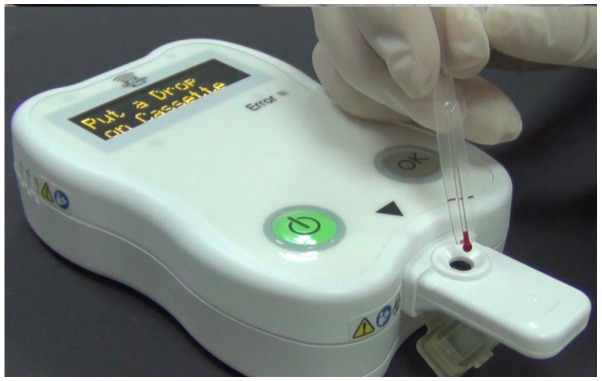
Physical photograph of the iProtin immunoassay and SH-SAW sensor chip coated with SARS-CoV-2 S protein.

**Figure 3 biosensors-12-00599-f003:**
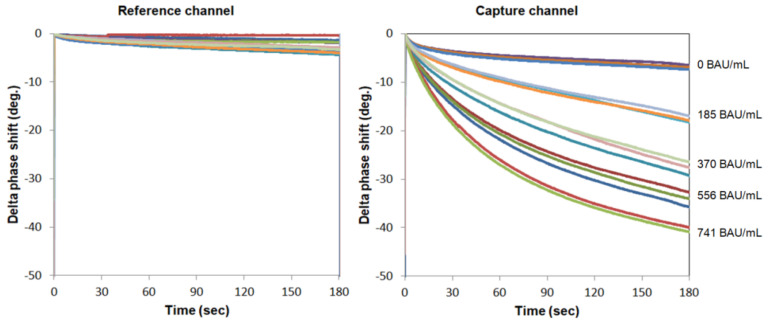
Real-time measurement of various concentrations of total anti-S protein antibodies in the reference and capture channels.

**Figure 4 biosensors-12-00599-f004:**
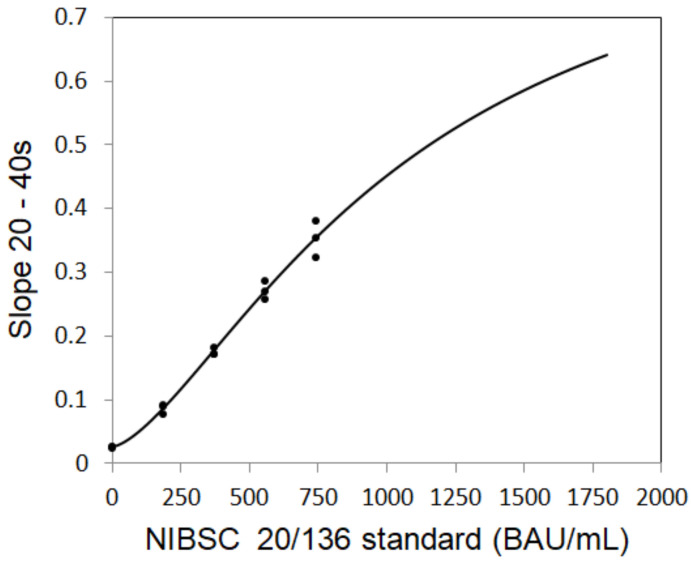
Establishment of four parameter logistics (4PL) standard curve.

**Figure 5 biosensors-12-00599-f005:**
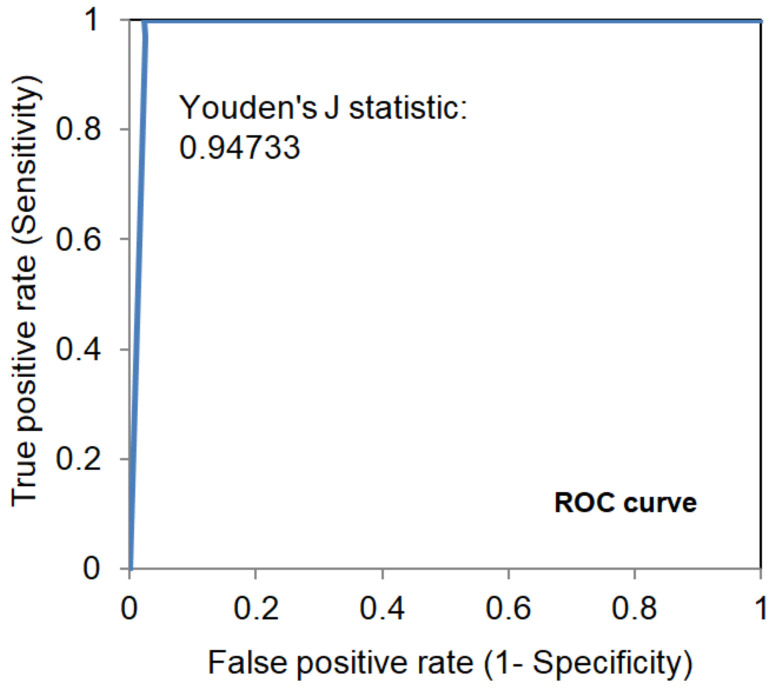
Receiver operating characteristic (ROC) analysis of S protein coated SAW biosensors.

**Figure 6 biosensors-12-00599-f006:**
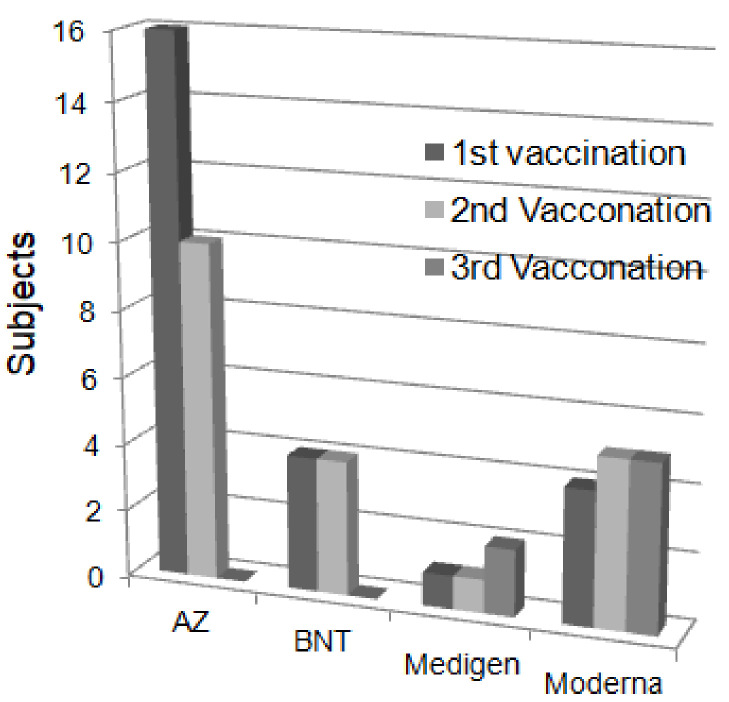
Distribution of vaccine brands received by subjects.

**Figure 7 biosensors-12-00599-f007:**
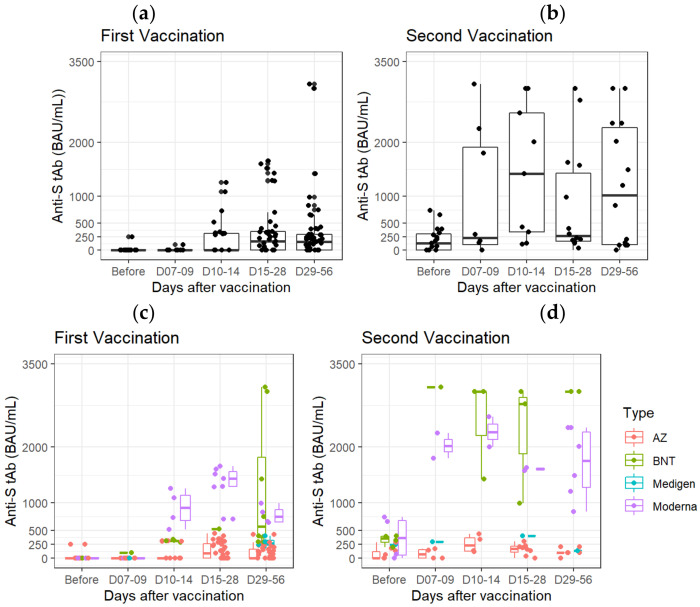
Follow-up and comparison of total anti-SARS-CoV-2 S protein antibody concentrations after the first (**a**,**c**) and second (**b**,**d**) vaccination. (**a**,**b**) All brands of vaccines; (**c**,**d**) different brands of vaccines.

**Figure 8 biosensors-12-00599-f008:**
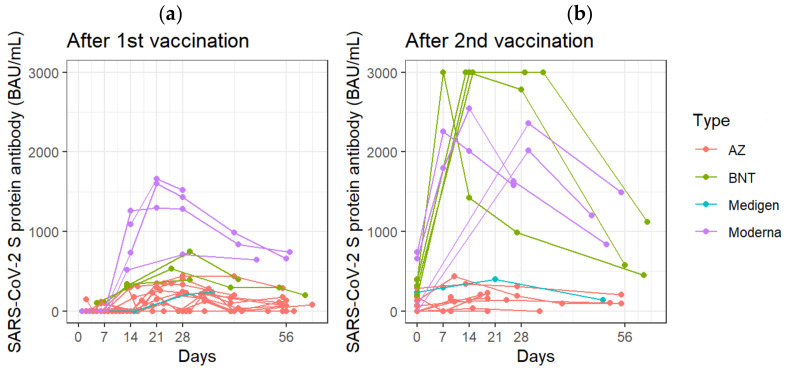
Follow-up measurements of anti-SARS-CoV-2 S protein total antibodies (**a**) at the indicated days after the first vaccination (**b**) at the indicated days after the second vaccination. Each line represents the concentration of antibodies in the blood continuously measured after receiving one brand of vaccine on the specified date.

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
