# Peer review of "Measurements of Anti-SARS-CoV-2 Antibody Levels after Vaccination Using a SH-SAW Biosensor"

_biosensors, 2022, doi:10.3390/bios12080599_

Round 1

Reviewer 1 Report

The present study reports on the detection of anti-SARS-CoV-2 antibody levels in whole blood after the first, second and third vaccination by using an SH-SAW biosensor. The latter is functionalized with SARS-CoV-2 trimeric S protein as binding partner for the anti-SARS-CoV-2 antibodies.

Unfortunately, the manuscript has many shortcomings as will be listed below:

1) the title is misleading as a) there are no “kinetic” measurements and b) the SH-SAW biosensor is well known and by far not “novel” (see also Ref. 13)

2) in lines 95, 125, 150, 172, 182, 194, and 217 errors are indicated (Error! Reference source not found), which make it hard or even impossible to follow the text

3) Except Figure 3, no Figure is mentioned in the text. The reader must find out by himself which figure regards to which text passage. Again, this makes it hard or even impossible to follow the text

4) line 54: please mention the interfering factors, which may lead to false positive results

5) line 64: please explain what 4PL stands for

6) line 111: the sensor chip is coated but not the proteins

7) line 121: please explain BAU/mL

8) line 133: what is a “schematic photo”?

9) section 2.1: Manufacturer’s locations are missing

10) line 24 and 165: LOD or LoD?

11) Figure 3: Can you explain the difference in delta phase shift for reference channel and capture channel at 0 BAU/mL?

12) Figure 5: there is no information in the manuscript how the true positive rate and the false positive rate have been determined.

13) Section 3.3: the number of subjects (1. Vaccination: 25; 2. Vaccination: 20; 3. Vaccination: 7) is too low to perform good statistics. Moreover, it is not clear who became which vaccination. It is not apparent with which vaccine the first, second and third vaccinations were carried out. That is, AZ + AZ, or AZ + BNT, or BNT + Moderna + Moderna, etc.

14) line 211 to 213: this statement is not supported by any data presented. If you refer to Fig. 8, you should mention this, although Fig. 8 does not support this anyway.

15) Figures 7 and 8: no a) and b) indicated in figures

16) line 223: “SH-SAW biosensor measures the phase shifts of the output signal based on binding event on the sensing region” – what exactly is sensed by SH-SAW? Is it only the bound antibody or the antibody with the surrounding water?

17) line 231:” We have confirmed that a 40-second measurement time is good enough and have performed a number of experiments using a 40-second measurement time.” How have you confirmed that the measurement time of 40 seconds is good enough? The signal is changing very strongly from 0 to 180 seconds. There is no objective criterium given, which supports to 40-second measurement time.

18) line 235: “The results showed that total anti-SARS-CoV-2 S protein antibodies spiked on days 10 to 14 after the first dose and on days 7 to 9 after the second dose.” Which data support this statement? In Fig. 8, many spikes are shown. For 2nd vaccination, most spikes are beyond 10 days (see Fig. 8b)

19) line 242: “The results of this study also suggested that there are individual differences in the antibody response to vaccination. Potential factors are age, gender, and genetics.” There are NO data in the present study shown concerning individual differences! This is pure speculation.

20) line 259: If the sample can be saliva, why has not saliva been used for the study instead of finger blood? Saliva is much easier to donate than blood

21) paragraph starting at line 290 should be moved to the Conclusion section

22) line303: the present study deals solely with anti-S antibodies. No data on anti-N antibodies are provided

Reviewer 2 Report

The present manuscript deals with the study of kinetics of anti-SARS-CoV-2 antibody levels after vaccination using a novel SH-SAW biosensor. The manuscript is interesting, it will contribute few new things to the literature. In my opinion, it should be revised.

Comments,

1) In the introduction, recent related works should be discussed, ref. Chemical Engineering Journal 430, 132966, 2022.

2) 2.1 Materials.. 2.4, 3.1, 3.2, 3.3, 3.4 Error! Reference source not found..??

3) 3.5. Kinetics of total anti-SARS-CoV-2 S protein antibodies after vaccination should be discussed in detail.

4) Figure 7 is not clear.

5) Conclusions should be rewritten.

6) References section should be checked carefully.

7) There are some typo and English errors and it should be rectified.

Round 2

Reviewer 1 Report

The manuscript has been significantly improved

Reviewer 2 Report

It can be accepted.